# First Report on *Leptospira* Species Isolated from Patients in Slovenia

**DOI:** 10.3390/microorganisms11112739

**Published:** 2023-11-09

**Authors:** Eva Ružić-Sabljić, Daša Podgoršek, Katja Strašek Smrdel, Andraž Celar Šturm, Mateja Logar, Andrea Pavlović, Tatjana Remec, Zvonko Baklan, Emil Pal, Tjaša Cerar Kišek

**Affiliations:** 1Institute of Microbiology and Immunology and Catedra for Microbiology and Immunology, Faculty of Medicine, University of Ljubljana, Zaloška 4, 1000 Ljubljana, Slovenia; eva.ruzic-sabljic@mf.uni-lj.si (E.R.-S.); dasa.podgorsek@gmail.com (D.P.); katja.strasek@mf.uni-lj.si (K.S.S.); andraz.celar-sturm@mf.uni-lj.si (A.C.Š.); 2Department of Pathology and Cytology, General Hospital Celje, Oblakova 5, 3000 Celje, Slovenia; 3Department of Infectious Diseases, University Medical Center, Zaloška 2, 1000 Ljubljana, Slovenia; mateja.logar@kclj.si; 4Catedra for Infectious Diseases and Epidemiology, Faculty of Medicine, University of Ljubljana, 1000 Ljubljana, Slovenia; 5Department of Infectious Diseases, General Hospital Celje, Oblakova 5, 3000 Celje, Slovenia; andreapavlov1@gmail.com; 6Department of Infectious Diseases, General Hospital Novo Mesto, Šmihelska Cesta 1, 8000 Novo Mesto, Slovenia; remect@sb-nm.si; 7Department of Infectious Diseases, University Medical Center Maribor, Ljubljanska Ulica 5, 2000 Maribor, Slovenia; zvonko.baklan@ukc-mb.si; 8Department of Infectious Diseases, General Hospital Murska Sobota, Ulica dr. Vrbnjaka 6, 9000 Murska Sobota, Slovenia; emil.pal@sb-ms.si; 9National Laboratory of Health, Environment and Food, Prvomajska 1, 2000 Maribor, Slovenia

**Keywords:** *Leptospira* sp., leptospirosis, Slovenia, *lfb* gene, MALDI-TOF, NotI-RFLP, MLST, WGS

## Abstract

Leptospirosis is an important worldwide zoonosis, and it has also been reported in Slovenia. The cultivation of Leptospira from human material is difficult. Despite that, we successfully isolated 12 human *Leptospira* strains isolated from patients between 2002 and 2020 and used various methods for the phenotypic and genotypic characterization of the strains, including matrix-assisted laser desorption/ionization time-of-flight mass spectrometry (MALDI-TOF) using our own MALDI-TOF data library, melting temperature analysis of the amplified *lfb1* gene, determination of *Leptospira* serogroups using rabbit immune sera, *NotI*-RFLP of the whole *Leptospira* genome, multilocus sequence typing (MLST) of seven housekeeping genes, and whole-genome sequencing (WGS)-based typing. We confirmed the presence of four pathogenic *Leptospira* species (*L. kirschneri*, *L. interrogans*, *L. borgpetersenii*, and *L. santarosai*) and three serogroups: Grippotyphosa, Icterohaemorrhagiae, and Sejroe. MALDI-TOF identified three of seven isolates at the species level and four isolates at the genus level. Serovars of 8 of the 10 strains were determined using *NotI*-RFLP. MLST showed that the clinical isolates belonged to sequence types ST17, ST110, and ST155. WGS confirmed the analysis of *Leptospira* strains using conventional methods. In addition, WGS provided better taxonomic resolution for isolate DDA 10944/10.

## 1. Introduction

Leptospirosis is the most widespread zoonosis, and it is present in Slovenia [1,2,3]. Recently, based on genotypic classification, it was proposed to divide the genus *Leptospira* into two clades, P and S, and four subclades, P1, P2, S1, and S2 [4]. Subclade P1 includes the 19 formerly pathogenic strains of *Leptospira* spp., including 8 human pathogenic strains (*Leptospira interrogans*, *Leptospira borgpetersenii*, *Leptospira kirschneri*, *Leptospira alexanderi*, *Leptospira noguchii*, *Leptospira santarosai*, and *Leptospira weilli*), subclade P2 contains strains with formerly intermediate pathogenic status, and clade S contains saprophytic strains.

Pathogenic *Leptospira* can infect many species of domestic and free-living animals, which excrete the bacteria through urine, contaminating fields, meadows, and bodies of water, in which *Leptospira* can survive for several weeks or even months. Human infection usually occurs through contact with a contaminated environment or infected animals [5,6].

Leptospirosis was first reported in Slovenia in 1938 in Murska Sobota County, which for decades was the Slovenian region where leptospirosis was endemic [1]. Leptospirosis occurs in several Slovenian regions and has an average 10-year incidence of 0.89 per 100,000 (2010–2020), which is one of the highest incidence rates in Europe [7].

*Leptospira* are fastidious bacteria, requiring specialized media for isolation and cultivation as well as long incubation times (up to several weeks) [8]. The successful cultivation of *Leptospira* requires the collection of specimens from a clinically well-diagnosed patient and immediate inoculation into a special medium, followed by transport of the sample to the laboratory for further processing. In Slovenia, this technique has been available to clinicians for many years. The first human *Leptospira* strain from Slovenia was isolated in 2002 from a young athlete who died after hemorrhagic pneumonia. Between 2002 and 2020, our laboratory successfully isolated 12 human *Leptospira* strains. The aim of the present study was to analyze the phenotypic and genotypic characteristics of the isolated *Leptospira* strains using both traditional and modern methods.

## 2. Materials and Methods

### 2.1. Leptospira Clinical Isolates

This study was conducted in accordance with the principles of the Declaration of Helsinki. The study was approved by the National Medical Ethics Committee of the Republic of Slovenia (No. 122/05/12).

All human *Leptospira* strains were isolated from blood drawn from patients on the first day of their hospitalization. The blood was immediately inoculated into tubes filled with Korthoff’s medium or, if not available, into hemoculture bottles, and they were sent to the laboratory. At the laboratory, the hemoculture flasks were reinoculated into three or four tubes of fresh Korthoff’s medium. The tubes were incubated at 28 °C and examined weekly for the presence of spirochetes for up to 12 weeks [9,10]. The isolated strains were stored in liquid nitrogen in the laboratory collection of the Institute of Microbiology and Immunology, Faculty of Medicine, Ljubljana, until further analysis. Table 1 shows the clinical isolates and the year of isolation.

In addition to the clinical isolates, we analyzed reference strains from the *Leptospira* sp. laboratory collection of the Laboratory for Diagnosis of Borreliosis and Leptospirosis at the Institute for Microbiology and Immunology of the Faculty of Medicine in Ljubljana (Table 2), some of which are routinely used for microagglutination tests (MATs). Reference isolates were used to monitor the performance of the methods.

### 2.2. Leptospira Serogroup Determination

The isolated human *Leptospira* strains were assigned to serogroups on the basis of rabbit immune sera obtained from the Royal Tropical Institute, Amsterdam (Appendix A). Strains were incubated with rabbit immune sera at a dilution of 1:400. After 45 min of incubation at 37 °C, samples were examined under a dark-field microscope. Samples with positive reactions were further diluted to 1:12,800, and the highest agglutination titer was determined. The unknown *Leptospira* strain isolated from the clinical material was assigned to a serogroup on the basis of the highest agglutination titer.

### 2.3. Leptospira Serovar Identification

Whole-genome restriction with the restriction enzyme *NotI* (*NotI*-RFLP) was used to identify serovars of the isolated human *Leptospira* strains. A large volume of the growing culture was sedimented and washed before being mixed with agarose and poured into molds. Whole agarose blocks were lysed (24 h at 37 °C) and digested (72 h at 50 °C) in buffer containing lysozyme (1 mg/mL), RNase (10 μg/mL), and proteinase K (0.5 mg/mL). After extensive washing, DNA was digested with the restriction enzyme *NotI* (20 E/200 μL, 24 h at 37 °C), and pulsed-field gel electrophoresis (PFGE) was performed. Restricted DNA fragments were separated for 24 h with ramping pulse times from 5 to 70 s; a molecular size marker of 50 to 1000 kb (Sigma, St. Louis, MO, USA) was included in each electrophoresis. Gels were stained with ethidium bromide solution for 30 min and imaged using the Bio-Rad Gel Doc™ XR system (Bio-Rad Technologies, Hercules, CA, USA). The *NotI*-RFLP patterns of the isolated *Leptospira* strains were compared to the restriction patterns of our reference strain of a known serovar [11,12,13]. Fingerprint patterns were analyzed by pairwise comparisons using the Dice coefficient. The mean values of the coefficients were used in the unweighted pair group method with arithmetic mean (UPGMA) using BioNumerics 7.1 software (Applied Maths, Austin, TX, USA) to generate a dendrogram with band comparison settings of 1.5% optimization and 1.5% tolerance of the band-position.

### 2.4. Leptospira Identification Using MALDI-TOF MS

The identification of isolated human *Leptospira* strains using matrix-assisted laser desorption/ionization time-of-flight mass spectrometry (MALDI-TOF MS) was performed using an in-house library constructed from the mass spectra of the reference *Leptospira* strains. For this purpose, each *Leptospira* culture was washed twice: once with phosphate buffer and once with absolute ethanol. The resulting precipitate was resuspended in formic acid and acetonitrile, mixed, centrifuged, and stored at −20 °C. The matrix and bacterial standard were prepared according to the manufacturer’s instructions (Bruker Daltonics, Bremen, Germany). We applied 1 µL of the extracted culture or bacterial standard to the MALDI carrier, allowed it to dry, added 1 µL of the matrix, and allowed it to dry again. Each standard was measured three times. For each reference strain, we obtained at least 24 images of high-quality mass spectra suitable for further processing. The obtained spectra were processed using FlexAnalysis version 3.0 and Biotyper OC-MSP software version 3.1 (Bruker Daltonics, Germany). The selected mass spectra were labeled with the name of the corresponding reference strain and entered into the data library. The quality of the entered data was checked by re-analyzing the reference strains entered into the data library. The results were displayed numerically (scores) and interpreted according to the level of identification reliability suggested by the manufacturer (scores >2 indicate identification of the species; scores between 1.7 and 1.9 indicate identification of the genus; and scores <1.7 indicate no identification). The mass spectra of both reference strains and clinical isolates were imported into BioNumerics 7.1 software, and a phylogenetic tree was drawn using the UPGMA method [14,15,16,17].

### 2.5. DNA Extraction

Genomic DNA was extracted using the QIAamp DNA Mini Kit (Qiagen, Hilden, Germany) or DNeasy blood and tissue kit (Qiagen, Hilden, Germany).

### 2.6. Leptospira Species Identification Using PCR-Tm

Species identification was conducted according to the method described by Mérien et al. [18]. Using primers LFB1-F and LFB1-R, we amplified a 331 bp fragment of the *lfb1* gene (locus LA0322), whose sequence polymorphism allowed for species identification and determined the melting temperature (Tm) of the amplicon (PCR-Tm). We performed several analyses for a given strain and compared its Tm with the Tm of the reference strains of *Leptospira* according to the instructions of Mérien et al. [18].

### 2.7. Leptospira Sequence Typing

For multilocus sequence typing (MLST), seven housekeeping genes (*glmU*, *pntA*, *sucA*, *caiB*, *tpiA*, *pfkB*, and *mreA*) were amplified according to the protocol described by Boonsilp et al. [19]. PCR products were detected by 2% agarose gel electrophoresis and sequenced in both directions. The sequencing reaction was performed using the BigDye Terminator v3.1 kit (Applied Biosystems, Waltham, MA, USA) on an ABI 3500XL genetic analyzer (Applied Biosystems). Sequences were analyzed using CLC Main Workbench 6.9.1, MLST Module (CLC Bio, Aarhus, Denmark), and allele numbers and sequence types (ST) were assigned.

### 2.8. Whole-Genome Sequencing

Whole-genome sequencing was performed on 9 isolates because we were unable to reculture the remaining 3 isolates from storage in liquid nitrogen. Genomic libraries were prepared using the Nextera XT Library Preparation Kit (Illumina, San Diego, CA, USA). Isolates were sequenced on the NextSeq 550 System (Illumina) using 2 × 149 bp paired-end reads chemistry. Fastp v0.23.2 [20] was used to trim the raw reads from the adapter sequences and low-quality reads. The quality of both raw and trimmed reads was assessed using FastQC v0.11.9 [21]. The assembly of trimmed reads into contigs was conducted with SPAdes v3.15.3 [22] using the default Kmer values and the “--careful” parameters. Quast v5.2.0 [23] was used for an assessment of the quality of the assemblies. Genome assemblies were annotated with prokka v1.14.6 [24] using the *Leptospira interrogans* strain FDAARGOS_203 as the reference genome. Species identification of the assembled reads was performed using the KmerFinder web application available at https://cge.food.dtu.dk/services/KmerFinder-3.2 (software v. 3.0.2, database v. 2022-07-11, accessed on 9 October 2023) [25,26,27].

For isolate DDA 10944/10, we additionally classified the reads using Kraken2 v2.1.3 [28]. We extracted *Leptospira santarosai*, *Leptospira noguchii*, and *Leptospira interrogans* reads using KrakenTools v.1.2 [29] and assembled reads separately using SPAdes v3.15.3 [22].

The sequence types of the isolates were determined in silico using MLST scheme #1 and MLST scheme #2 from the *Leptospira* spp. database on PubMLST [30].

### 2.9. Phylogenetic Analysis

Phylogenetic analysis of the isolates was performed in three different ways. The *lfb1* and *secY* genes were extracted from annotated genome assemblies of 27 isolates (9 Slovenian and 18 from the NCBI reference database genome) of different *Leptospira* species and serovars. The online tool Mafft [31] was used for multiple sequence alignment of the above genes using the “L-INS-i” method. Phylogenetic trees based on nucleotide differences between genes were constructed using the unweighted pair group method with an arithmetic mean (UPGMA) and visualized using the online tool iTOL [32]. A phylogenetic tree based on core genome MLST (cgMLST) was constructed by importing the allele data from the Institut Pasteur *Leptospira* database into Seqsphere+ (Ridom) and creating a neighbor- joining tree (NJT) based on the allelic differences between the isolates.

### 2.10. Data Availability

The generated raw data, assembled genomes, and *secY* and *lfb1* nucleotide sequences were submitted to the European Nucleotide Archive (ENA). The assigned accession numbers are PRJEB67337 for the study, OY754500-OY754507 for the *secY* nucleotide sequences, and OY754508-OY754515 for the *lfb1* nucleotide sequences. The sequences were deposited in the MLST Database (http://pubmlst.org/leptospira/ accessed on 2 November 2023) under ID 1774-1781.

## 3. Results

Using different typing methods, we identified the phenotypic and genotypic characteristics of human *Leptospira* strains isolated from patients with leptospirosis in Slovenia between 2002 and 2020. For the two isolates from 2017 to 2020, only serogroup identification and WGS analysis were performed; for three isolates (KB 185/02, VV 206/02, and PZ 9474/06), WGS analysis was not possible because we could not reculture the isolates from liquid nitrogen.

### 3.1. Leptospira Serogroup Determination

Serogroup determination with rabbit immune sera was performed for the 12 isolated human *Leptospira* strains and the 15 reference strains (Appendix A). All 15 reference strains were identified according to the expected serogroup. Using the same hyperimmune sera, we successfully identified the serogroups of all the clinical isolates. Five clinical isolates were assigned to the Grippotyphosa serogroup, three to the Icterohaemorrhagiae serogroup, three to the Sejroe serogroup, and one to the Bataviae serogroup (Table 3).

### 3.2. Leptospira Serovar Identification

We analyzed 10/12 isolated human *Leptospira* strains and 15 reference *Leptospira* strains and constructed a UPGMA dendrogram based on the *NotI*-RFLP results (Figure 1). *NotI*-RFLP band patterns were determined for 12 of the 15 reference strains. For two reference strains, Tarassovi Tarassovi Mitis Johnson and Ballum Castellonis Castellon 3, we obtained identical *NotI*-RFLP banding patterns, whereas we did not obtain *NotI*-RFLP for the reference strain Semaranga Patoc Patoc 1. The reference strains were thus divided into 12 different *NotI*-RFLP band patterns representing 12 different serovars. Figure 1 shows that the reference strains Pyrogenes Pyrogenes Salinem, Tarassovi Tarassovi Mitis Johnson, and Ballum Castellonis Castellon 3 are the most closely related, with a 94 match percentage, followed by Panama Panama CZ 214 K and Javanica Javanica Veldart Batavia with a 90 match percentage.

The clinical isolates were defined based on the *NotI*-RFLP of the reference strains. Five of the ten clinical isolates (FF 291/02, KB 185/02, VV 206/02, PZ 9474/06, and SS 8049/14) had *NotI*-RFLP banding patterns identical to those of Grippotyphosa Grippotyphosa Moskva V (match percentage 90.4); three of the isolates (ČP 226/02, SS 166/02, and OD 8720/11) had an 87.5 match percentage to Icterohaemorrhagiae Copenhageni Wijnberg; and two clinical isolates (MJ 112721/08 and DDA 10944/10) had distinct *NotI*-RFLP patterns that did not match any of the strains tested. Strain MJ 112721/08 had already been placed in the Sejroe serogroup because it was well defined with the *NotI*-RFLP pattern (Figure 1). Isolate DDA 10944/10, which had previously been assigned to the serogroup Bataviae, had a low match percentage with the reference serovar Bataviae and was closer to the reference serovar Sejroe (64.5 match percentage) according to the *NotI*-RFLP (Figure 1). The isolates LA 9525/17 and PM 155/20 were not analyzed.

### 3.3. Leptospira Identification Using MALDI-TOF MS

The MALDI-TOF library data do not contain the mass spectra of *Leptospira*. Therefore, we had to create our own library by inputting the mass spectra of 21 reference strains of *Leptospira* from our collection. The mass spectra were recorded according to the library protocol of Bruker Daltonics. The protein profiles (e.g., mass spectra) of the *Leptospira* reference strains were specific and did not match any of the more than 4000 existing profile files in the Bruker data library.

To determine whether the newly created spectra were useful and appropriate for typing clinical isolates, we used the reference strains with which we created the library as the test samples. We were able to correctly identify 12 of 14 reference strains, all of which scored greater than 2.300, suggesting reliable species-level identification. For the remaining two reference strains (Canicola Canicola Hond Utrecht IV and Cynopteri Cynopteri 3522C), we obtained scores for two different species. *L. interrogans* was the most abundant species identified in the reference strains (Appendix A).

We then analyzed seven clinical isolates using MALDI-TOF mass spectrometry (Table 3). We were unable to reculture the remaining isolates from storage in liquid nitrogen. Using MALDI-TOF MS, we identified three of the seven available clinical isolates to the species level and four of the seven isolates to the genus level (Table 3).

Based on the obtained mass spectra, a cluster analysis was performed, and a spanning tree was constructed from the reference strains and clinical isolates. The clinical isolates from the Slovenian patients are related, but strain DDA 10944/10 differs from the other clinical isolates (Figure 2).

### 3.4. Leptospira Species Identification Using PCR-Tm

PCR-Tm correctly identified the *Leptospira* species of 14 of the 15 reference strains, indicating that the method used was appropriate for the identification of the isolated human *Leptospira* strains. The PCR-Tm analyses of the reference strains are listed in Appendix A and are comparable to the melting temperatures reported in the literature. We were unable to amplify DNA from the saprophytic strain Semaranga Patoc Patoc 1 because the oligonucleotide primers used for species identification are specific for pathogenic *Leptospira* [18].

Using PCR-Tm, 10/12 clinical isolates were identified to species: 5 were identified as *L. kirschneri*, 3 as *L. interrogans*, and 2 as *L. borgpetersenii* (Table 3).

### 3.5. Leptospira Sequence Typing

Sequence types (STs) based on MLST scheme #1 were successfully determined for eleven of the twelve clinical isolates and twelve of the fifteen reference strains. Using the standard method, amplification and Sanger sequencing of housekeeping genes, we were able to determine the STs for eight isolates. For two of the clinical isolates, ST could not be determined because the combination of allele numbers did not match any of the combinations in the MLST database. MLST scheme #1, that we used for typing, did not contain the combination of alleles and STs that would identify the Semaranga Patoc Patoc 1, Sejroe Sejroe M84, and Tarrasovi Tarassovi Mitis Johnson strains, so we could not identify these three reference strains.

Using WGS and in silico MLST scheme #1, we were able to determine the STs in eight isolates, the results of the standard approach and WGS agreed in five isolates, and we were able to determine the STs of strain MJ 112721/08 and strains LA 9525/17 and PM 155/20, for which the standard approach was not performed. Using both approaches, we were able to determine the STs in 11/12 strains: ST17, ST110, and ST155 in 3, 5, and 3 strains, respectively (Table 3 and Appendix A).

We identified 12 different sequence types in the reference strains (Appendix A).

### 3.6. WGS

The quality metrics, KmerFinder results, STs by MLST scheme 1 and 2, and cgMLST results are shown in Appendix A. Figure 3 shows the neighbor-joining tree (NJT) based on the allelic differences between the isolates.

We succeeded in typing all strains, except for DDA 10944/10. Classification of the reads by Kraken 2 revealed that three *Leptospira* species were present: *L. santarosai*, *L. noguchii*, and *L. interrogans*. The highest number of fragments, 8,221,729, was directly assigned to *L. santarosai* (Appendix A). The numbers were much lower for *L. noguchii* and *L*. *interrogans*. Using KrakenTools v.1.2 [29], we extracted the reads assigned to taxa 28,183 (*L. santarosai*), 28,182 (*L. noguchii*), and 173 (*L. interrogans)* and assembled the reads using SPAdes v3.15.3 [22]. We were only able to assemble the genome of *L. santarosai*.

We performed an MLST analysis in the PubMLST database to define STs according to schemes #1 and #2. The results of MLST #1 scheme agreed to the conventional MLST #1 scheme by Boonsilp et al. [19]. The analysis of the isolate DD 10944/10 presented no results in the PubMLST database, probably due to the reasons described above (mixed culture/isolates) (Appendix A).

cgMLST analysis is based on the 545 genes that are highly conserved across the genus *Leptospira* according to Guglielmini et al. in the database of Institut Pasteur [33]. The majority of the genes were successfully called, and cgST was defined (Appendix A). The analysis of some isolates (SS 166/02, OD 8720/11, SS 8049/14) returned multiple cgSTs with a 100% or 99.6% match. After careful manual analysis of each locus from isolates with multiple cgSTs with loci from the Institut Pasteur database, we determined one cgST per isolate with a 100% match over all loci, that is, cgST 199 for both problematic isolates (Appendix A). Again, we could not determine cgST for the isolate DD 10944/10, probably due to the same reasons as mentioned above. The locus matched in the Institut Pasteur database was low, at 76.9% (419/545).

Additionally, we extracted the sequences of genes *secY* and *lfb1* from both isolate’s and reference strain’s genomes and performed a phylogenetic analysis of the selected genomes. Both analyses confirmed the cgMLST phylogenetic tree and classified our isolates among *Leptospira* strains: *L. borgpetersenii* (ML 11721/08, LA 9525/17, and PM 155/20), *L. kirschneri* (FF 291/02 and SS8049/14), *L. interrogans* (CP 226/02, OD 8720/11, and SS 166/02), and *L. santarosai* (DDA 10944/10) (Figure 4). *L. biflexa* is not included in the *lfb1* phylogenetic tree as this saprophytic species does not have this gene.

For some isolates (KB 185/02, VV 206/02, and PZ 9474/06), WGS analysis was not possible as we could not reculture them from liquid nitrogen.

## 4. Discussion

Using various traditional and modern typing methods, we analyzed the phenotypic and genotypic characteristics of *Leptospira* strains isolated from patients with leptospirosis in Slovenia between 2002 and 2020.

This is the first report of *Leptospira* strains isolated from patients in Slovenia. Although the typing of *Leptospira* is not clinically relevant, it is crucial for public health and epidemiological studies [34]. There are several typing methods that differ in cost and require a lot of time, expertise, and equipment, making some suitable only for reference laboratories. We aimed to determine whether the results of the phenotypic and genotypic typing methods we used were consistent, whether these methods could represent *Leptospira* diversity, and whether the epidemiology of *Leptospira* strains isolated from patients in Slovenia was comparable to the epidemiology of European *Leptospira*.

Serotyping is commonly used in routine diagnostics to confirm the agglutination properties of strains listed in MAT (Table 3) and is useful for identifying new *Leptospira* isolates. Five of our twelve clinical isolates were assigned to the Grippotyphosa serogroup, and three to the Icterohaemorrhagiae serogroup, the most widely distributed European serogroups. Of the remaining four clinical isolates, three were assigned to the Sejroe serogroup and one to the Bataviae serogroup (Table 3). Problems in identifying *Leptospira* using the serotyping method may occur if the isolate belongs to a serogroup whose immune serum is not included in the test or if the strain loses its agglutination properties [34].

*Leptospira* serovars were determined by *NotI*-RFLP for the clinical isolates and the reference strains. The serovar of the reference strain Semaranga Patoc Patoc 1 could not be determined because this strain tends to self-degrade, making the DNA unsuitable for *NotI*-RFLP analysis. Most other studies focused on pathogenic *Leptospira* strains, and *NotI*-RFLP was not performed for saprophytic *Leptospira* strains [13,35], although Naigowit et al. [36] described a characteristic *NotI*-RFLP of a Patoc 1 strain. The *NotI*-RFLP patterns for two of the reference strains (Tarassovi Tarassovi Mitis Johnson and Ballum Castellonis Castelon 3) were identical, suggesting identical serovars. These strains both belong to the species *L. borgpetersenii*, but we expected them to have different *NotI*-RFLP patterns because they are reference strains from the European *Leptospira* reference laboratory (Figure 1). It is possible that the strains were mixed in the original cultures and that multiple passages resulted in one strain overgrowing the other, or that multiple reinoculations resulted in the strains mixing during laboratory work [13].

According to *NotI*-RFLP, eight clinical isolates were assigned to the serogroups Grippotyphosa (FF 291/02, SS 8049/14, KB 185/02, VV 206/02, and PZ 9474/06) or Icterohaemorrhagiae (OD 8720/11, ČP 226/02, and SS 166/02) (Table 3, Figure 1). The serovars identified for our clinical isolates are comparable to those described in other studies, with minor variations likely due to different electrophoresis conditions [12,13,14,34].

Serotyping using immune sera is a faster typing method than *NotI*-RFLP, but the possibility of cross-agglutination makes the interpretation of results subjective and difficult [36]. The *NotI*-RFLP method is thorough but time-consuming and carries the additional risk that DNA in agarose blocks may be damaged, resulting in the loss of results [36,37].

*Leptospira* strains can be identified to the species level using a variety of methods, including PCR-Tm, MALDI-TOF mass spectrometry, MLST, and WGS-based analyses [15,18,19,38,39,40,41,42,43]. PCR-Tm analysis is considered to be a simple and rapid method for the identification of *Leptospira* [18]. In Slovenian patients, *L. kirschneri* and *L. interrogans*, the most widespread species in Europe and worldwide, predominated (Appendix A). Although the PCR amplicon of *L. interrogans*, endemic to Europe, has a very similar melting temperature to that of *L. noguchii*, endemic to South America, we defined our clinical isolates as *L. interrogans* [18]. The majority of the reference strains were classified as *L. interrogans*, followed by *L. kirschneri* and *L. borgpetersenii*, and a single reference strain was defined as *L. noguchii* (Appendix A). Mérien et al. [18] reported the same melting temperature for *L. kirschneri* and *L. noguchii*, but we were able to distinguish these species because the average melting temperatures differed by 0.2 °C.

MALDI-TOF has recently become an established technique for bacterial identification, but the library MALDI-TOF does not yet contain data for *Leptospira*. Therefore, we created our own data library. The data library was created using a procedure similar to that described by Girault [44]. The adequacy of the imported data was verified by using it to identify the 14 reference strains from MAT, whose mass spectra had been previously imported into the library. Of these strains, 12 were correctly identified. For the remaining two strains, two different species with very similar score values were identified. We hypothesize that these two strains were either contaminated with other *Leptospira* species or with other bacteria that interfered with the mass spectral identification. Using MALDI-TOF, we identified three of eight clinical isolates to the species level: two as *L. interrogans* and one as *L. borgpetersenii*. The score value for the other five isolates was lower and only allowed for reliable identification to the genus level. To identify all the clinical isolates to species level, we would need to create a better library with a larger number of reference strains. Other authors have also described MALDI-TOF MS for the identification of *Leptospira*, and it is possible to extend this method with ClinPro Tools software version 3.0 to allow for the identification of *Leptospira* up to the serovar level [14,15,16,17]. This method could be useful in the future to identify *Leptospira* protein profiles and analyze possible relationships with virulence factors. The MALDI-TOF mass spectrometry method is easy to use and allows for extremely rapid identification, with results available within minutes. The major drawback is that the data library for *Leptospira* is not commercially available and must be created by each user. In addition, identification problems can occur when samples have low *Leptospira* concentrations, which is often the case for *Leptospira* because the strains generally grow poorly.

Due to the high genetic diversity of *Leptospira*, which limits the use of a single MLST scheme, there are several different schemes encompassing six or seven different loci [40,41,42]. In this study, we used two approaches, the standard MLST of Boonsilp et al. [19] and the WGS-based MLST and cgMLST. MLST data are collected in an online database, the PubMLST *Leptospira* database, which allows for a quick and easy analysis and comparison between laboratories, as was the case for our strains. We found three Leptospira STs in isolates from Slovenian patients: ST17, ST110, and ST155 in three, five, and three patients, respectively (Table 3). Currently, there are 1580 isolates in the PubMLST Leptospira database (accessed 20 October 2023). Of the 1580 isolates, 1142 were determined using MLST scheme #1 at ST. The most common ST is ST17 (138/1142), followed by ST34 (126/1142) and ST149 (83/1142). The sequence types ST110 and ST155 represent a minority with 16 and 13 isolates, respectively. Of the 1580 isolates in the database, only 77 isolates are from Europe, of which 58 were determined using MLST scheme #1 ST. The most common STs are ST155 (9 isolates), ST 17 (5 isolates), ST 24 (4 isolates), ST110 (4 isolates), and ST149 (4 isolates). In total, 20 European isolates are of human origin, of which 17 were determined using MLST scheme #1 ST, with ST17, which was determined in 4 isolates, being the most common ST.

We have used several methods to determine the species, serotype, and/or serovar of the genus *Leptospira*. The main differences between the methods are the accuracy with which the method determines the characteristics of the strain and the technique used. We were not able to determine all the studied characteristics of all Slovenian isolates, but the species-level agreement is high for those we were able to determine. The most problematic was the isolate DD 10944/10, where there were many possible interpretations of the results: PCR-Tm identified *L. borgpetersenii*, but WGS, also a molecular method, determined that most of the data belonged to *L. santarosai*, although reads of *L. interrogans* and *L. noguchii* were also present, but in much smaller numbers. This is probably why the interpretation of the MLST schemes (conventional and WGS) was not possible, why MALDI-TOF could not correctly identify *L. santarosai*, and why *NotI*-RFLP showed a different pattern. The patient with this isolate travelled across many continents, so the exact site of infection (and a possible strain or strains of *Leptospira*) could not be determined.

It would be interesting to perform SNP analysis of WGS genomes from isolates of Slovenian patients in the future from different regions of Slovenia. Although the results of the different methods used are in agreement regarding the level of *Leptospira* strain, there might be some differences at the nucleotide level. In addition, it would be interesting to analyze virulence genes to compare Slovenian isolates with isolates from other parts of the world.

## 5. Conclusions

We confirmed the presence of the three main pathogenic *Leptospira* species in Slovenia, *L. kirschneri*, *L. interrogans*, and *L. borgpetersenii*, and the three main serogroups, Grippotyphosa, Icterohaemorrhagiae, and Sejroe. Simple identification methods (serogrouping and MALDI-TOF) are useful for routine laboratory work, but more detailed information on *Leptospira* strains can be obtained with more complex and time-consuming typing methods, preferably with WGS.

## Figures and Tables

**Figure 1 microorganisms-11-02739-f001:**
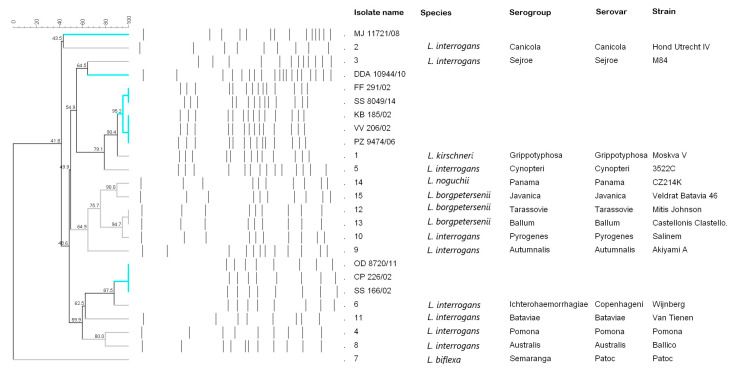
Cluster analysis and *NotI*-RFLP banding patterns of *Leptospira* strains isolated from patients (cyan) in Slovenia and the reference *Leptospira* strains (grey). A UPGMA dendrogram using a dice similarity coefficient with a 1.5% band position tolerance was constructed for the 10 clinical and 15 reference *Leptospira* strains using BioNumerics 7.1 software.

**Figure 2 microorganisms-11-02739-f002:**
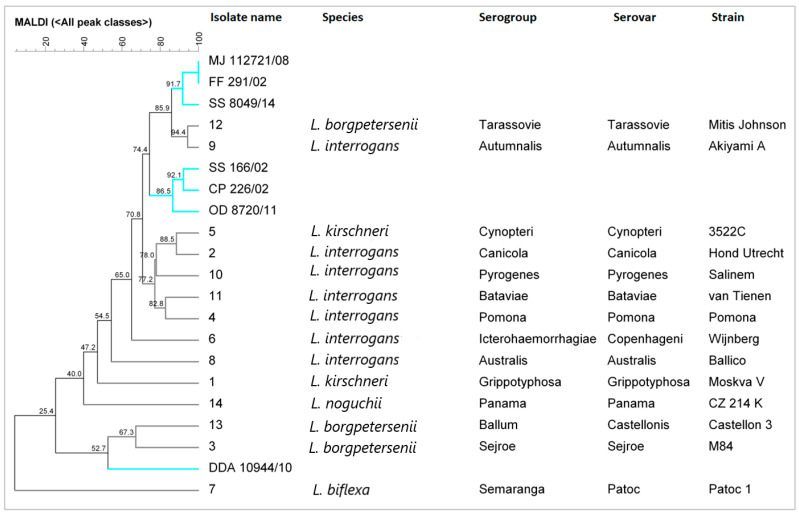
Cluster analysis of *Leptospira* strains isolated from patients in Slovenia (cyan) and the reference *Leptospira* strains (black) using MALDI-TOF mass spectrometry.

**Figure 3 microorganisms-11-02739-f003:**
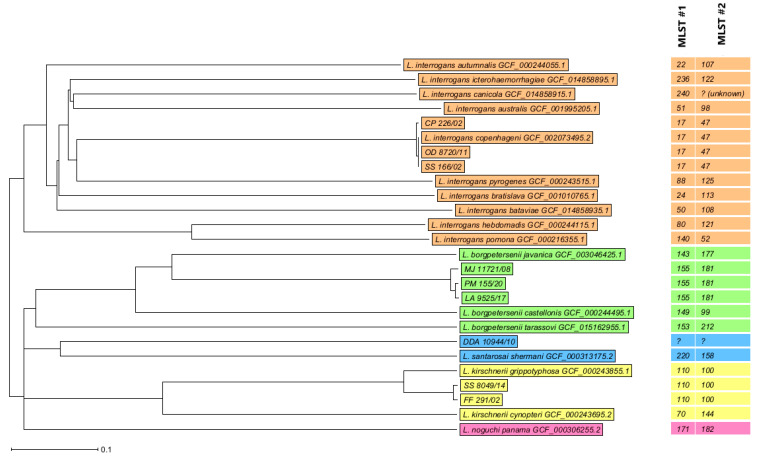
Neighbor-joining tree based on 500 columns, no missing values. Distance based on columns form *Leptospira* MLST 1 (6), MLST 2 (5), *Leptospira* cgMLST (489). Orange: *L. interrogans*, green: *L. borgpetersenii*, blue: *L. santarosai*, yellow: *L. kirschnerii*, magenta: *L. noguchi*.

**Figure 4 microorganisms-11-02739-f004:**
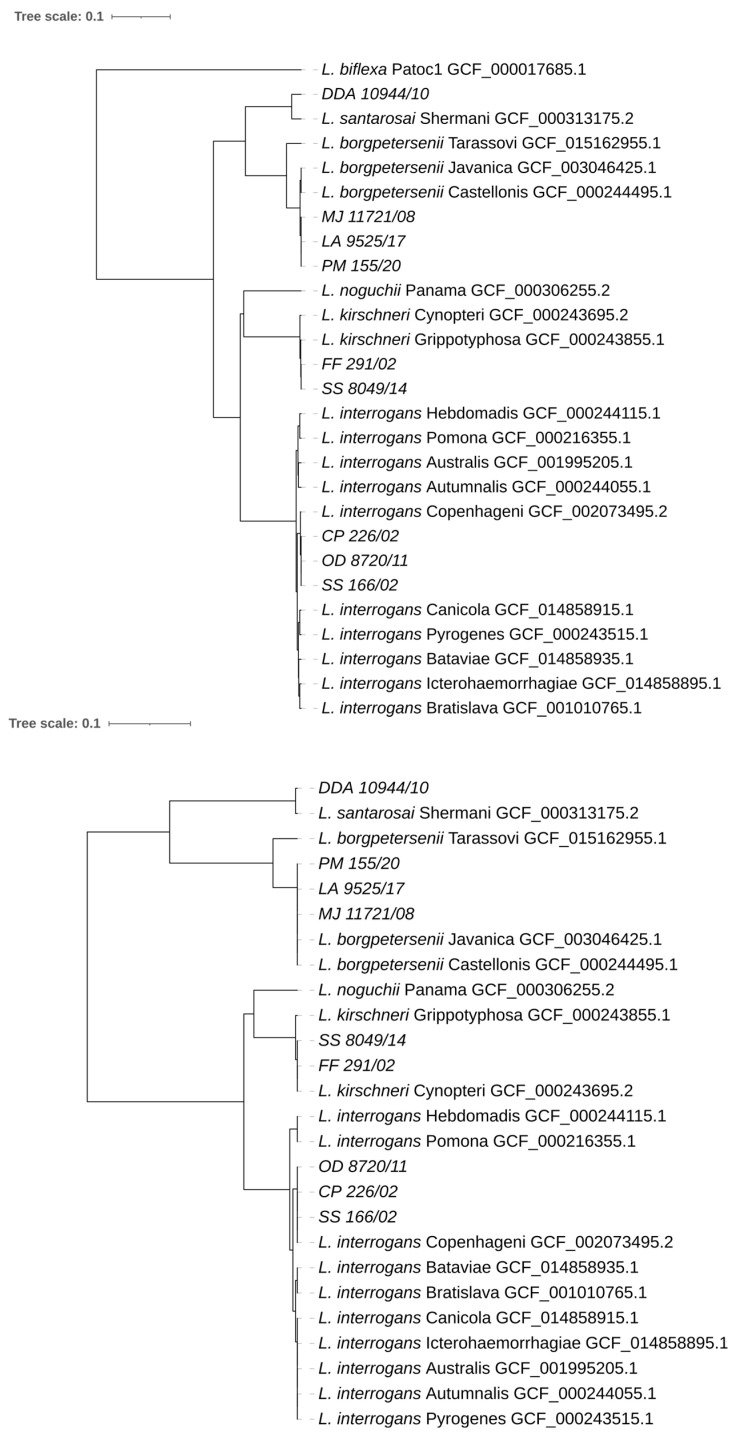
Phylogenetic trees based on nucleotide differences between the *secY* (**upper**) and *lfb1* (**lower**) genes were constructed using the unweighted pair group method with arithmetic mean (UPGMA). The size of a fragment of secY was 1383 bp; the outgroup is *L. biflexa* Patoc1 (acc. No. GCF_000017685.1). The size of a fragment of lfb1 ranges from 363 bp to 372 bp; *L. biflexa* has no *lfb1* gene.

**Table 1 microorganisms-11-02739-t001:** *Leptospira* sp. isolated from blood of patients in Slovenia between 2002 and 2020.

No.	Isolate Identification	Year of Isolation
1	SS 166/02	2002
2	KB 185/02	2002
3	VV 206/02	2002
4	FF 291/02	2002
5	ČP 226/02	2002
6	PZ 9474/06	2006
7	MJ 112721/08	2008
8	DDA 10944/10	2010
9	OD 8720/11	2011
10	SS 8049/14	2014
11	LA 9525/17	2017
12	PM 155/20	2020

**Table 2 microorganisms-11-02739-t002:** *Leptospira* strains from the laboratory collection used as reference strains.

No.	Species	Serological Group	Serovar	Strain Name	Origin
1	*L. interrogans*	Canicola	Canicola	Hond Utrecht IV ^1^	A
2	*L. interrogans*	Pomona	Pomona	Pomona ^1^	A
3	*L. interrogans*	Icterohaemorrhagiae	Copenhageni	Wijnberg ^1^	LJ
4	*L. interrogans*	Australis	Australis	Ballico ^1^	A
5	*L. interrogans*	Autumnalis	Autumnalis	Akiyami A ^1^	A
6	*L. interrogans*	Pyrogenes	Pyrogenes	Salinem ^1^	A
7	*L. interrogans*	Bataviae	Bataviae	Van Tienen ^1^	P
8	*L. interrogans*	Icterohaemorrhagiae	Icterohaemorrhagiae	RGA	A
9	*L. interrogans*	Australis	Bratislava	Jez Bratislava	P
10	*L. interrogans*	Hebdomadis	Hebdomadis	Hebdomadis	LJ
11	*L. borgpetersenii*	Sejroe	Sejroe	M84 ^1^	P
12	*L. borgpetersenii*	Tarrassovi	Tarassovi	Mitis Johnson ^1^	LJ
13	*L. borgpetersenii*	Ballum	Castellanis	Castellon 3 ^1^	A
14	*L. borgpetersenii*	Javanica	Javanica	Veldart Batavia 46 ^1^	A
15	*L. kirschneri*	Cynopteri	Cynopteri	3522C ^1^	A
16	*L. kirschneri*	Grippothyphosa	Grippotyphosa	Moskva V ^1^	A
17	*L. santarosai*	Shermani	Shermani	1342K	A
18	*L. noguchii*	Panama	Panama	CZ 214 K ^1^	A
19	*L. noguchii*	Pomona	Proechiniys	1161 U	P
20	*L. alexanderi*	Hebdomadis	Manzhuang	A23	P
21	*L. biflexa*	Semaranga	Patoc	Patoc 1 ^1^	A, P, ZG, LJ

^1^ Strains used in routine laboratory work for microagglutination tests. LJ = Faculty of Veterinary Medicine, Ljubljana; P = Institut Pasteur, Paris, France; A = Royal Tropical Institute, Amsterdam; ZG = Faculty of Veterinary Medicine, Zagreb.

**Table 3 microorganisms-11-02739-t003:** Comparison of results of different typing approaches for species identification of *Leptospira* strains isolated from patients in Slovenia.

							WGS			
No.	Isolate No.	PCR-Tm Identification	Serotyping Identification	*NotI* I RFLP	MALDI-TOF Identification	MLST 1 ^8^	MLST 1 ^9^	Kmer Finder	*secY*	*lfb1*
1	SS 166/02	*L. interrogans*	Icterohaemorrhagiae	87.5% match to ICW ^6^	*Leptospira* sp. (*L. interrogans*) ^4^	17	17	*L. interrogans* serovar Copenhageni str. FDAARGOS	*L. interrogans*	*L. interrogans*
2	KB 185/02	*L. kirschneri*	Grippotyphosa	90.4% match to GGMV ^7^	ND ^5^	110	ND ^5^	ND ^5^	ND ^5^	ND ^5^
3	VV 206/02	*L. kirschneri*	Grippotyphosa	90.4% match to GGMV ^7^	ND ^5^	110	ND ^5^	ND ^5^	ND ^5^	ND ^5^
4	FF 291/02	*L. kirschneri*	Grippotyphosa	90.4% match to GGMV ^7^	*Leptospira* sp. (*L. interrogans*) ^4^	110	110	*L. kirschneri* strain FMAS_PN5	*L. kirschneri*	*L. kirschneri*
5	ČP 226/02	*L. interrogans*	Icterohaemorrhagiae	87.5% match to ICW ^6^	*L. interrogans* ^3^	17	17	*L. interrogans* serovar Copenhageni str. FDAARGOS	*L. interrogans*	*L. interrogans*
6	PZ 9474/06	*L. kirschneri*	Grippotyphosa	90.4% match to GGMV ^7^	ND ^5^	110	ND ^5^	ND ^5^	ND ^5^	ND ^5^
7	MJ 11721/08	*L. borgpetersenii*	Sejroe	Distinct pattern	*L. borgpetersenii* ^2^	-	155	*L. borgpetersenii* FMAS_AP4	*L. borgpetersenii*	*L. borgpetersenii*
8	DDA 10944/10 ^1^	*L. borgpetersenii*	Bataviae	Distinct pattern	*Leptospira* sp. (*L. santarosai*) ^4^	ND	ND	*L. santarosai* serovar Shermani str. LT 821	*L. santarosai*	*L. santarosai*
9	OD 8720/11	*L. interrogans*	Icterohaemorrhagiae	87.5% match to ICW ^6^	*L. interrogans* ^1^	17	17	*L. interrogans* serovar Copenhageni str. FDAARGOS	*L. interrogans*	*L. interrogans*
10	SS 8049/14	*L. kirschneri*	Grippotyphosa	90.4% match to GGMV ^7^	*Leptospira* sp. (*L. kirschneri*) ^4^	110	110	*L. kirschneri* strain FMAS_PN5	*L. kirschneri*	*L. kirschneri*
11	LA 9525/17	ND	Sejroe	ND	ND	ND	155	*L. borgpetersenii* FMAS_AP4	*L. borgpetersenii*	*L. borgpetersenii*
12	PM 155/20	ND	Sejroe	ND	ND	ND	155	*L. borgpetersenii* FMAS_AP4	*L. borgpetersenii*	*L. borgpetersenii*

^1^ Isolate of a South American patient who had contracted *Leptospira* while travelling in Asia. The patient became ill and was hospitalized in Slovenia. ^2^ Highly probable species identification.^3^ Secure genus identification; probable species identification. ^4^ Probable genus identification.^5^ Unsuccessful recultivation from liquid nitrogen. ^6^ Icterohaemorrhagiae Copenhageni Wijnberg. ^7^ Grippotyphosa Grippotyphosa Moskva V. ^8^ Amplification and Sanger sequencing according to Boonsilp et al. ^9^ WGS-based MLST typing scheme 1 by Boonsilp et al. ND—not done.

## Data Availability

All the relevant data are included in the manuscript in aggregated format.

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
