# Peer review of "First Report on Leptospira Species Isolated from Patients in Slovenia"

_microorganisms, 2023, doi:10.3390/microorganisms11112739_

Round 1
Reviewer 1 Report
Comments and Suggestions for Authors
The authors are to be commended for their efforts in analyzing Leptospira isolates from Slovenia. The results are interesting, especially since there are few studies published regarding leptospirosis in eastern european countries. Molecular epidemiology for Leptospira, which are fastidious bacteria to grow, is lagging behind other bacteria and therefore, I think this paper makes an important contribution to the field and provides a valuable model and example of the kind of analyses that can be performed for the identification of circulating isolates.
Comments :
Please update the taxonomy of the genus Leptospira ; there are now at least 69 described Leptospira species divided in P1, P2, S1 and S2 subclades (see Vincent et al. 2019).
Avoid the use of « sensu stricto », this designation of Leptospira interrogans is no more valid.
If available, please add informations on the sample date, how many days after the onset of symptoms ?
Tables (4, 6, 8 and 10) showing the data of reference strains can be moved to the supplementary data
Strains DDA 19944/10 and MJ 112721/08 are not fully identified using the different methods. As mentioned by the authors, this may be due to contamination or mixed infections. Reisolation of the culture onto solid medium should enable to obtain clonal cultures (from a single colony).
For PFGE, authors can also compare the restriction profiles with other published NotI-restriction profiles (see for example, references Mende et al. Am J Trop Med Hyg 2013 89(2):380-4 and Galloway et al. Am J Trop Med Hyg. 2008;78(4):628-32)
Figure 2 is not very informative
MALDI-Tof has already been used for the identification of Leptospira, please cite the following references and discuss your results (libraries should already contain Leptospira data).
Girault et al. Methods Mol Biol. 2020; 2134:23-29
Sonthayanon et al. . PLoS Negl Trop Dis. 2019 ;13(4):e0007232.
Karcher et al. Braz J Microbiol. 2018 ;49(4):900-908
I would like to see the authors expanding their discussion to consider the public health relevance of these results. For example, any idea what could be the animal reservoir of Grippotyphosa ? clinical and epidemiological data of patients infected with Grippotyphosa, etc
Comments on the Quality of English LanguageMost common grammar mistakes are the major issue in the current script. Needs to be revised.
Author Response
We would like to thank you very much for taking the time to review this manuscript and for the constructive criticism, which has allowed a substantial improvement of our article. You can find the detailed responses below.
The authors are to be commended for their efforts in analyzing Leptospira isolates from Slovenia. The results are interesting, especially since there are few studies published regarding leptospirosis in eastern european countries. Molecular epidemiology for Leptospira, which are fastidious bacteria to grow, is lagging behind other bacteria and therefore, I think this paper makes an important contribution to the field and provides a valuable model and example of the kind of analyses that can be performed for the identification of circulating isolates.
Comments :
Please update the taxonomy of the genus Leptospira ; there are now at least 69 described Leptospira species divided in P1, P2, S1 and S2 subclades (see Vincent et al. 2019).
Response: Thank you for your comment. Done as suggested.
Avoid the use of « sensu stricto », this designation of Leptospira interrogans is no more valid.
Response: Thank you. We have removed all ''sensu stricto'' designations.
If available, please add informations on the sample date, how many days after the onset of symptoms ?
Response: We agree that this kind of information would be very interesting, but unfortunately we were not able to get it.
Tables (4, 6, 8 and 10) showing the data of reference strains can be moved to the supplementary data.
Response: Thank you for your comment. Done as suggested.
Strains DDA 19944/10 and MJ 112721/08 are not fully identified using the different methods. As mentioned by the authors, this may be due to contamination or mixed infections. Reisolation of the culture onto solid medium should enable to obtain clonal cultures (from a single colony).
Response: Thank you for your suggestion. We have decided to do WGS instead since reculturing isolates from liquid nitrogen takes quite a long time and is not always successful.
For PFGE, authors can also compare the restriction profiles with other published NotI-restriction profiles (see for example, references Mende et al. Am J Trop Med Hyg 2013 89(2):380-4 and Galloway et al. Am J Trop Med Hyg. 2008;78(4):628-32)
Response: Thank you for your comment. Done as suggested.
Figure 2 is not very informative
Response: Thank you for your comment. Figure 2 was removed.
MALDI-Tof has already been used for the identification of Leptospira, please cite the following references and discuss your results (libraries should already contain Leptospira data).
Girault et al. Methods Mol Biol. 2020; 2134:23-29
Sonthayanon et al. . PLoS Negl Trop Dis. 2019 ;13(4):e0007232.
Karcher et al. Braz J Microbiol. 2018 ;49(4):900-908
Response: Thank you for your comment. Suggested references were cited.
I would like to see the authors expanding their discussion to consider the public health relevance of these results. For example, any idea what could be the animal reservoir of Grippotyphosa ? clinical and epidemiological data of patients infected with Grippotyphosa, etc
Response: We agree that discussion would be beneficial from a public health perspective but we do not believe we have enough data to allow such a discussion.
Comments on the Quality of English Language
Most common grammar mistakes are the major issue in the current script. Needs to be revised.
Response: We have revised the manuscript.
Reviewer 2 Report
Comments and Suggestions for Authors
Please find attached.

Comments on the Quality of English LanguageI strongly suggest that authors submit the manuscript for rigorous review by a native English speaker.
Author Response
We sincerely appreciate your time in reviewing this manuscript and your helpful criticism, both of which have made a significant enhancement to our paper. The full responses are listed below.
In the study “First report on the Leptospira species isolated from patients in Slovenia” the authors performed serological and genetic characterization of leptospires isolated from humans. The study is relevant, since obtaining isolates is a difficult task and genetic and serological data are very welcome in the literature and can contribute to further studies. However, in its current scope, the study presents important limitations that, once properly adapted, could make the manuscript suitable for publications. Please find below some of my main considerations:
Firstly, I strongly suggest that authors submit the manuscript for rigorous review by a native English speaker, after making all necessary changes.
Please carefully review, from the title to the references, including captions and figures, the italicized writing “Leptospira”. In some places the name is in italics, but in many it is not.
Response: Thank you for your comment. Corrected.
The purpose of the study is unclear. Do the authors want to characterize isolates from human patients, or do they want to evaluate the sensitivity/specificity of different diagnostic methods? I was able to clearly see two different studies: One focusing on the standardization and comparison of methods and the other applying the methods to characterize the isolates. And this made the presentation of the study very confusing. Wouldn’t it be interesting for this study to be divided into two articles? One using the reference strains and focused on the development, standardization and comparison between characterization methods and the other focused only on the isolates from human patients, using previously evaluated and standardized methodologies? The authors created their own data library and evaluated whether MALDI-TOF could identify the reference strains to the species level. That would be enough to write an individual article! And then they genetically characterized pathogenic leptospires for the first time in Slovenia...that would be another article!
Some of the methodologies used are old and very laborious (e.g.: PCR-TM, RFLP), in addition to generating inconclusive results. I believe that there are currently more modern and efficient methodologies for the genetic characterization of leptospires.
Response: We agree that the purpose of the study was not very clear and that the methods were a bit out of the data. In the revised manuscript, we have included WGS-based analysis and focused more on isolate characterization. We hope that this has resulted in a clearer objective for the study. We are happy to revise the manuscript again if necessary.
The authors performed genetic sequencing of 7 housekeeping genes for the MLST approach, but did not sequence the lfb-1 gene, which could quickly determine the Leptospira species. Furthermore, I strongly encourage the amplification and sequencing of the secY gene, currently the most widely used genetic marker for Leptospira, with excellent taxonomic resolution for species definition and haplogroup identification.
The dendrograms constructed based on the NotI-RFLP and MALDI-TOF (Figures 1 and 4) analyzes are not conclusive. It is not possible to observe well-defined clades, and species are mixed within the same cluster. Phylogenetic trees and haplotype networks based on the lfb-1 and secY genes would be more efficient not only for a clear observation of species identification, but also for a better view of molecular epidemiology (based on hosts and geographic locations, for example).
Response: Thank you very much for your suggestion. We extracted the sequences of the secY and lfb1 genes from both the isolate and reference strain genomes and performed a phylogenetic analysis of the selected genomes.
Based on this, I consider a study using only well-standardized methodologies for serological and genetic characterization of pathogenic leptospires to be more appropriate, and I strongly encourage a more robust analysis of molecular epidemiology, for example, based on the MLST approach, lfb-1 and secY genes, the authors can compare the relationship between the patients' strains and strains from other hosts and geographic regions, being able to make important inferences regarding origin and distribution.
The sequences of the 7 MLST genes must be deposited in GenBank and the accession numbers provided.
Response: The generated raw data and assembled genomes were submitted to the European Nucleotide Archive for the study under accession number PRJEB67337.
Regarding the ethics committee, this should be the first topic of the material and methods.
Response: Done as suggested.
Why did the authors opt for a molecular approach to defining serovars? Why didn't they perform a serological technique?
Response: We appreciate your feedback. The primary justification for not utilizing a serological methodology is that our lab has long employed NotI-RFLP as a means of defining serovars.
The resolution of the figures must be substantially improved.
Response: Done.
Scientific names must be in italics.
Response: Corrected.
In Figure 1, the column titles are missing. The colors of the legend and the tree do not match in Figure 4 .
Response: Column titles were added in Figure 1 , for Figure 4 legend was substituted with description.
Table 3 can be included as supplementary material
Response: Done as suggested.
I have other small observations to make in the text, but I really prefer to see a first answer to these major questions, and then I am available to provide a second review of the study.
Round 2
Reviewer 2 Report
Comments and Suggestions for Authors
Attached.

Comments on the Quality of English LanguageAuthor Response
We would like to thank you very much for taking the time to review this manuscript for the second time and for the constructive criticism, which again has significantly contributed to improving our manuscript. Please find our detailed responses below.
Title: Please remove “the” before “Leptospira”
Response: Thank you for your comment. Done as suggested.
Abstract:
Line 23: “Leptospirosis is a worldwide disease, a zoonosis, that has also occurred in Slovenia”. I suggest to rewrite: “Leptospirosis is an important worldwide zoonosis, that is also reported in Slovenia”
Response: Thank you for your suggestion. Done as suggested.
Line 24: “Cultivation of Leptospira from human material is difficult. We successfully cultured 12 human Leptospira ….” I suggest to rewrite: “Cultivation of Leptospira from human material is difficult. Despite that, we successfully isolated 12 human….”
Response: Thank you for your suggestion. Done as suggested.
Line 31: remove “the” before “four”
Response: Removed as suggested.
Line 32: remove “the” before three
Response: Removed as suggested.
Line 36: “In addition, WGS provided better resolution”. In what sense? taxonomic? Species definition? Serogroup? Serovar? Strain?
Response: Thank you for your comment. We have added the adverb 'taxonomic'. The sentence now reads: ''In addition, WGS provided better taxonomic resolution for isolate DDA 10944/10''.
Introduction
Line 46: replace the comma with "and" before Leptospira weilli
Response: Replaced.
Lines 56-60: Make it clear that Leptospira is a fastidious spirochete and its culture and isolation is a challenge, requiring special media and long incubation time.
Response: We changed the text to make these points clearer.
Line 63: “our laboratory collected 12 human Leptospira isolates”. Did the authors collect the isolate from a culture bank or collect the sample and perform isolation? Make this clear, it's confusing.
Response: We changed the text to make it more clear.
Material and Methods
Table 1: If all biological samples are blood, put it in the legend and remove it from the table; “Isolate identification no.” should be only “Isolate identification”.
Response: Corrected as suggested.
Line 82: “In addition to the clinical isolates, we analyzed reference strains from our laboratory …”. With what objective this was done? How did this contribute to the main objective of the study that was identify and characterize Leptospira human isolates? Make it clear.
Response: We have included reference isolates to monitor performance of the methods. This information was also added to the manuscript.
Line 82 and Table 2: “our laboratory”. Please put the name of the laboratory and the respective culture collection.
Response: Done as suggested.
I suggest rearranging the subtopics, it's confusing in the current format: For example, DNA extraction has been used for various purposes, but has been cited several times.
Response: Thank you very much for your great suggestion. Subtopics were rearranged.
I suggest the authors list first the techniques that used the isolates (serogrouping, RFLP, MALDI-TOF) and then those that used the DNA (PCR-Tm, MLST, WGS). Furthermore, I suggest the authors insert all the phylogenetic analysis (based on WGS, secY and lfb-1) carried out in a final topic "Phylogenetic analysis".
Response: Thank you for your comment. Done as suggested.
Line 191: Data must also be deposited on the MLST platform and GenBank. I strongly suggest depositing the secY and lfb-1 gene sequences separately from the genome in GenBank. This will strongly contribute to further studies involving the molecular epidemiology of leptospires worldwide.
Response: Thank you for your suggestions. The sequences were submitted to the MLST platform and ENA. The following text was added to the manuscript: ''The generated raw data, assembled genomes, secY and lfb1 nucleotide sequences were submitted to the European Nucleotide Archive (ENA). The assigned accession numers are PRJEB67337 for the study, OY754500-OY754507 for the secY nucleotide sequences and OY754508-OY754515 for lfb1 nucleotide sequences The sequences were deposited in the MLST Database (http://pubmlst.org/leptospira/) under ID 1774-1781.''
Since all three databases, the DNA DataBank of Japan (DDBJ), the European Nucleotide Archive (ENA) and the GenBank at NCBI, exchange data on a daily basis, the data should be accessible in all databases.
Rearrange subtopics according to the same suggestion of Material and Methods section
Response: Done as suggested.
Line 201: This subtitle is confusing, please rewrite
Response: Rewritten.
Table 3. Define what is “ND”
Response: Definition added.
Please try to put all the scientific names of the figures in italics
Response: Thank you for your comment. We have put all the scientific names in figures in italics.
Figure 1 and 4. Please improve the resolution of these figures. Leave it the same way as Figure 2.
Response: Resolution was improved.
Figure 4. Please include the accession numbers of reference strains included in the analysis.
Response: Accession numbers of reference strains were added.
Include in the legend: the size of the fragment used for each gene (bp); indicate which sequence was used as outgroup.
Response: The requested information were added to the legend.